# Single-Center Study of Lymphoepithelioma-Like Carcinoma of Uterine Cervix over a 10-Year Period

**DOI:** 10.3390/medicina55120780

**Published:** 2019-12-09

**Authors:** Angel Yordanov, Martin Karamanliev, Milena Karcheva, Assia Konsoulova, Mariela Vasileva-Slaveva, Strahil Strashilov

**Affiliations:** 1Department of Gynecologic Oncology, Medical University Pleven, 5800 Pleven, Bulgaria; 2Department of Surgical Oncology, Medical University Pleven, 5800 Pleven, Bulgaria; martinkaramanliev@gmail.com; 3Department of Epidemiology, Medical University Pleven, 5800 Pleven, Bulgaria; milena_karcheva@abv.bg; 4Complex Oncological Center Burgas, 8000 Burgas, Bulgaria; dr.konsoulova@gmail.com; 5EXTRO-Lab, Department of Therapeutic Radiology and Oncology, Medical University of Innsbruck, 6020 Innsbruck, Austria; sscvasileva@gmail.com; 6Tyrolean Cancer Research Institute, 6020 Innsbruck, Austria; 7EORTC Pathobiology Group, 1200 Brussels, Belgium; 8Department of Plastic and Reconstructive Surgery, MU-Pleven, 5800 Pleven, Bulgaria; dr.strashilov@gmail.com

**Keywords:** lymphoepithelioma-like cervical cancer, overall survival rate, lymph node involvement, prognosis, follow-up

## Abstract

*Background and objectives:* Lymphoepithelioma-like carcinoma (LELC) is a histological type of malignant tumor arising from the uncontrolled mitosis of transformed cells originating in epithelial tissue. It is a rare subtype of squamous cell carcinoma of the uterine cervix. There are significant differences in frequency, mean age, viral status, and outcomes in Asian or Caucasian patients. *Materials and Methods:* A retrospective study of all cases of lymphoepithelioma-like carcinoma of the cervix at the Clinic of Oncogynecology, University Hospital, Pleven, Bulgaria between 1 January 2007 and 31 December 2016 was performed. All patients were followed-up till March 2019. We analyzed some clinical characteristics of the patients, calculated the frequency of lymphoepithelioma-like carcinoma of the cervix from all patients with stage I cervical cancer, and looked at the overall survival rate, the 5-year survival rate, and the correlation between overall survival, lymph node status, and the size of the tumor. *Results:* The frequency of lymphoepithelioma-like carcinoma was 3.3% for all cases with cervical carcinoma at stage I. The mean age of the patients with LELC was 49.6 years (range 32–67). Fourteen patients (82.4%) were in the FIGO IB1 stage, three patients (17.6%) were in the FIGO IB2 stage. Lymph nodes were metastatic in three patients (17.6%), non-metastatic in 13 patients (76.5%), and unknown in one patient. The overall survival rate was 76.47% for the study period and the 5-year survival rate of the patients that were followed-up until the 5th year (14 patients) was 69.23%. *Conclusions:* Lymphoepithelioma-like carcinoma is a rare SCC subtype, but it could be more frequent among western patients than previously thought. Our results do not confirm the data showing low risk of lymph metastasis and good prognosis of LELC, which is why we think that the treatment in these cases has to be more aggressive than is reported in the literature.

## 1. Introduction

Lymphoepithelioma-like carcinoma (LELC) is a histological type of malignant tumor arising from the uncontrolled mitosis of transformed cells originating in epithelial tissue. It is a common type of poorly differentiated epithelial cells in the nasopharynx [1,2,3]. LELC is seen in salivary glands, lungs, nasopharynx, skin, thymus, stomach, urinary bladder, and uterine cervix [2,4]. The diagnosis is pathomorphological.

According to World Health Organization, cervical cancer is the fourth most frequent cancer in women with an estimated 570,000 new cases in 2018 representing 6.6% of all female cancers, and approximately 90% of deaths from cervical cancer occur in low- and middle-income countries [5]. The most common histological type of cervical neoplasia is squamous cell carcinoma (SCC), at around 80% of all cases. A rare subtype of SCC is lymphoepithelioma-like carcinoma. It was reported for the first time by Hamazaki et al. in 1968 [6]. In the literature, LELC is described mainly in case reports [7].

There are significant differences in frequency, mean age, viral status, and outcomes in Asian or Caucasian patients [7,8].

The objective of our study was to analyze the frequency of LELC in hospitalized women with cervical cancer, as well as the clinical characteristics, treatments, and prognosis of LELC.

## 2. Materials and Methods

A retrospective study of all cases of LELC of the cervix at the Clinic of Oncogynecology, University Hospital, Pleven, Bulgaria between 1 January 2007 and 31 December 2016 was performed.

Clinical data were collected from patients’ medical records. Patients with clinical stage I who were initially referred to surgery were analyzed. All histological slides were reviewed by an expert, and the diagnosis was reconfirmed. Pathologic and clinical staging were performed according to TNM classification or FIGO. All patients were followed-up until March 2019. The follow-up was done at 3, 6, 9, 12, 15, 18, 21, and 24 months and then annually, including clinical examination, blood tests, and chest X-ray. Annually a whole-body contrast-enhanced CT was performed. We analyzed some clinical characteristics of the patients, calculated the share of LELC from all patients with stage I cervical cancer, and looked at the overall survival (OS) rate, the 5-year survival rate, the correlation between OS and lymph node status, and the correlation between OS and the size of the tumor. Statistical analysis was done by using SPSS for Windows.

## 3. Results

Six hundred and thirty patients with cervical cancer were operated on in our clinic during the study period. Seventeen of the women had LELC, which represented 3.3% of all cases with cervical carcinoma at stage I (all patients who were referred directly to surgery and no neoadjuvant treatment was performed) during the study period. All of the patients had a histological diagnosis before radical surgery (except for one case) from a cervical biopsy that was done due to abnormal genital bleeding. In one patient, the biopsy showed benign pathology but due to persistence of complaints, a laparohysterectomy (LHT) was performed.

The mean age of the patients with LELC was 49.6 years (range 32–67). In one patient a simple hysterectomy was performed because of benign histology after dilation and curettage. In all other patients a radical hysterectomy with total pelvic lymph node dissection was performed. In all patients, adjuvant radiotherapy was done. In 14 patients, immunohistochemical staining (IHC) for human papilloma virus (HPV) and Epstein–Barr virus (EBV) was done, and eight of them (47.1%) were positive for any or both viruses and six (35.3%) were negative for both viruses. In three patients (17.6%), the exam was not performed because of the lack of paraffin blocks.

Fourteen patients (82.4%) were in the FIGO IB1 stage, and three patients (17.6%) were in the FIGO IB2 stage. The size of the primary tumor was <2 cm in five patients (29.4%), 2–4 cm in nine patients (52.9%), and >4 cm in three patients (17.6%). Lymph nodes were metastatic in three patients (17.6%), non-metastatic in 13 patients (76.5%), and unknown in one patient.

The overall survival rate was 76.47% for the study period, and the 5-year survival rate of the patients that were followed-up until the 5th year (14 patients) was 69.23%.

When comparing the OS between the non-metastatic lymph node group and the metastatic lymph node group, there was a trend of a lower OS in the metastatic lymph node group (Figure 1), which did not reach statistical significance (*p* = 0.087).

When studying the correlation between the OS and the size of the tumor (Figure 2), there was no significant difference between the groups (*p* = 0.327).

Some clinical and pathoanatomical characteristics of the patients are presented in Table 1.

## 4. Discussion

Histological LELC is composed of poorly defined islands of undifferentiated cells in a background intensely infiltrated by lymphocytes. The tumor cells have uniform, vesicular nuclei with prominent nucleoli and moderate amounts of slightly eosinophilic cytoplasm. The cell borders are indistinct, often imparting a syncytial-like appearance to the groups. This typical microscopic appearance and immunohistochemistry for epithelial and lymphoid markers can help in differentiating cervical LELC from the poorly differentiated squamous cell carcinoma and lymphoproliferative lesions.

In the female genital tract, LELC has been reported in the vulva, vagina, uterine cervix, and endometrium [9].

When LELC affects the cervix, it is believed to have a better prognosis than the normal SCC of the cervix due to lack of lymph node metastasis [3,10]. Significant differences in the incidence of this type of carcinoma in Asian and Caucasian races have been reported. It represents 0.7% of all primary cervical malignancies among the Western population and is about 5.5% among the Asian population [3]. There is also a difference in the mean age of diagnosis in these patients. In Asian patients it is reported to be between 43 and 50 years (range 30–72) and mean age of 42.3 years in Western patients (range 21–58) [11]. It is assumed that LELC is associated with Epstein–Barr virus (EBV) infection in Asians, whereas Westerns are associated with human papillomavirus (HPV), or viral genesis cannot be proven [3,12,13].

Typically, the diagnosis is made at an early stage and there is no involvement of the lymph nodes. This could be the reason for better prognosis reporting for this disease [11].

In our study, we presented 17 cases with LELC of uterine cervix, which represented 3.3% from all stage I cases with cervical carcinoma during the study period. This rate was four times greater than the literature results. This frequency can be explained by the fact that only those patients who were FIGO I stage and were directly referred to surgery were included in this study. All other cases of cervical cancer were excluded. However, we believe that it is quite possible that the incidence of LELC in Western patients is higher than reported, as it is determined on the basis of case reports and small case series.

The mean age in our group was 49.6 (ranging from 32 to 67), which was slightly higher than the published data so far, although Martorell et al. found that the mean age of their patients was 69 years [11].

In our study, only three patients (17.6%) were diagnosed with a tumor larger than 4 cm in diameter, confirming the data from the world literature that LELC is diagnosed early. However, the fact that in nine cases the tumor size was between 2 and 4 cm indicates that the diagnosis was not performed at an early stage. This could be due to the health culture of the population and problems with the coverage of the screening program. All patients had a history of abnormal genital and contact bleeding for at least one year. In two of the patients that died, lymph metastases were observed. The rapid progression of the disease, which could be explained by its possible hematogenic dissemination, was noticeable. In all four patients that died, the tumor was less than 4 cm, in two cases it was less than 2 cm, and in two cases it was between 2 and 4 cm. There was no relationship between the size of the tumor and its prognosis. There was a trend in lower OS in the metastatic lymph node group, which did not reach statistical significance. This could be explained with the small number of patients in the study.

## 5. Conclusions

Lymphoepithelioma-like carcinoma is a rare SCC subtype, but it may be more frequent among western patients than previously thought. Our results do not confirm the data showing low risk of lymph metastasis and good prognosis of LELC, which is why we think that the treatment in these cases has to be more aggressive than is reported in the literature. Due to the low incidence of this disease, a lot is still unknown. Larger studies in the area are needed.

## Figures and Tables

**Figure 1 medicina-55-00780-f001:**
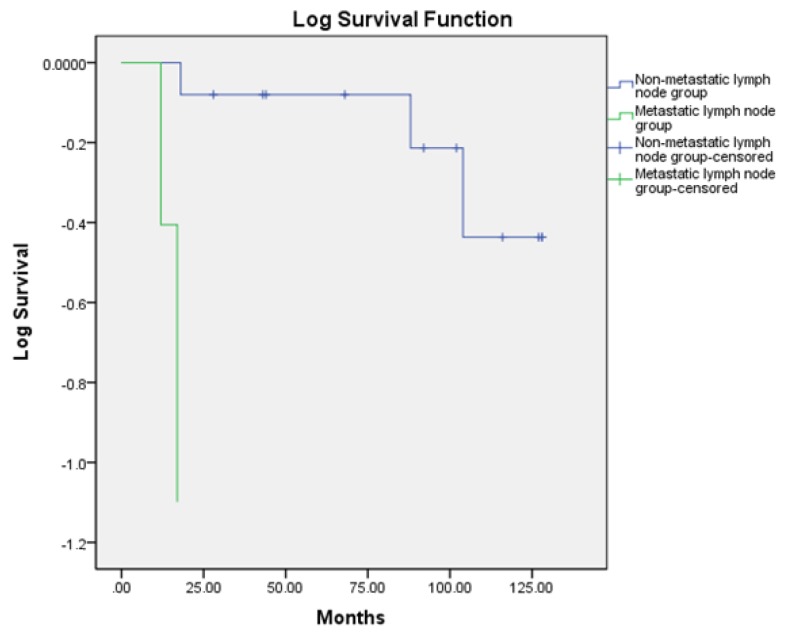
Comparing the OS between the non-metastatic lymph node group and metastatic lymph node group.

**Figure 2 medicina-55-00780-f002:**
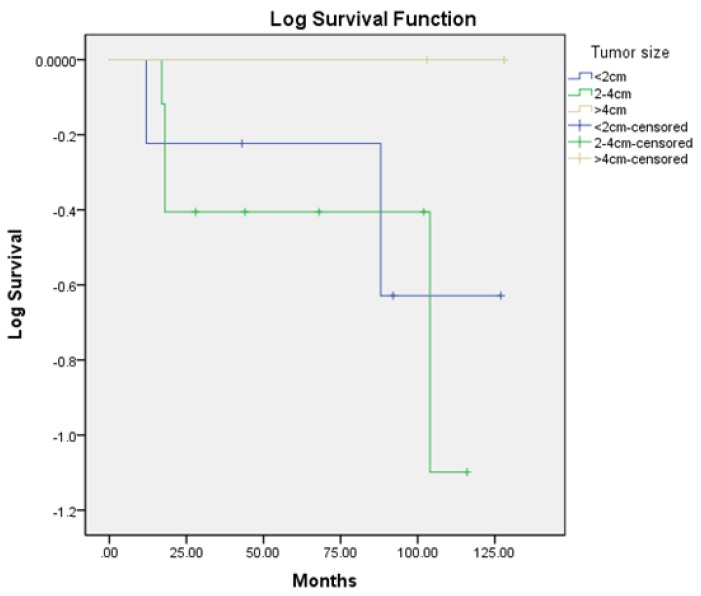
Comparing the OS in groups with different size of the tumor.

**Table 1 medicina-55-00780-t001:** Clinical and pathoanatomical characteristics of the patients.

Case	Age	Treatment	Clinical Stage	Tumor Size (cm)	Recurrence	Outcome
1	67	RH + PLND	pT1b1pN1Mo	b/n 2–4	Unknown	Died on 8th month
2	58	RH + PLND	pT1b1pNoMo	<2	Liver metastases	Died on 88th month
3	42	RH + PLND	pT1b1pNoMo	b/n 2–4	No	Alive on128th month
4	47	RH + PLND	pT1b2pN1Mo	>4	No	Alive on 128th month
5	48	RH + PLND	pT1b1pNoMo	b/n 2–4	No	Alive on 104th month
6	38	RH + PLND	T1b1pNoMo	<2	No	Alive on 127th month
7	46	RH + PLND	pT1b1pNoMo	b/n 2–4	No	Alive on 116th month
8	59	TH	pT1b2NoMo	>4	No	Alive on 103th month
9	49	RH + PLND	pT1bpNoMo	<2	No	Alive on 102th month
10	59	RH + PLND	pT1b1pNoMo	b/n 2–4	Bone metastases	Died on 18th month
11	40	RH + PLND	pT1b1pNoMo	b/n 2–4	No	Alive on 92th month
12	49	RH + PLND	pT1b1pN1Mo	b/n 2–4	Unknown	Died on 16th month
13	34	RH + PLND	pT1b1pNoMo	b/n 2–4	No	Alive on 68th month
14	66	RH + PLND	pT1b2pNoMo	>4	No	Alive on 52th month
15	61	RH + PLND	pT1b1pNoMo	b/n 2–4	No	Alive on 44th month
16	48	RH + PLND	pT1b1pNoMo	<2	No	Alive on 43th month
17	32	RH + PLND	T1b1pNoMo	<2	No	Alive on 28th month

RH—radical hysterectomy; TH—total hysterectomy; PLND—pelvic lymph node dissection.

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
