# Peer review of "Single-Center Study of Lymphoepithelioma-Like Carcinoma of Uterine Cervix over a 10-Year Period"

_medicina, 2019, doi:10.3390/medicina55120780_

Round 1

Reviewer 1 Report

The authors present a retrospective study concerning to LELC. Once these tumors are rare, I believe that the manuscript merits are the patient number and time expended to that analyses and follow up (3,3% out of 630 women - period since 2007 until 2016). Moreover, the manuscript brings information for clinical and researchers that could complement these tumors literature scarce. 

I suggest that author insert the viral data (HPV and/or EBV) of the patients in the results section. 

Author Response

Thank you for the review. We were performed IHC for both viruses and the results are inserted in the result section.

Reviewer 2 Report

This manuscript has interesting results, but needs minor revisions before it can be accepted for publication. 

1. Please add tumor size in Table 1.

2. Since LELC is rare subtype of uterine cancer and this study includes relatively larger number of cases, assessment of viral status (EBV and HPV) in this study will be valuable to characterize this rare entity in Caucasian patients.

3. In materials, please describe selection of stage I patients in this retrospective study and its reason

Author Response

Thank you for the review.

We added tumor size in table 1. We examined the viral status in these patients by IHC and we added them in the result section. We described and explained why only patients in clinical stage I were included in this research.